# Long COVID burden and risk factors in 10 UK longitudinal studies and electronic health records

Ellen J. Thompson [1,19✉], Dylan M. Williams [2,3,19✉], Alex J. Walker [4,19], Ruth E. Mitchell [5,6], Claire L. Niedzwiedz [7], Tiffany C. Yang[8], Charlotte F. Huggins[9], Alex S. F. Kwong [5,10], Richard J. Silverwood [11], Giorgio Di Gessa [12], Ruth C. E. Bowyer [1], Kate Northstone[6], Bo Hou[8], Michael J. Green [13], Brian Dodgeon[11], Katie J. Doores [14], Emma L. Duncan [1], Frances M. K. Williams [1], OpenSAFELY Collaborative*, Andrew Steptoe [12], David J. Porteous [9], Rosemary R. C. McEachan[8], Laurie Tomlinson [15], Ben Goldacre[4], Praveetha Patalay[2,11], George B. Ploubidis[11], Srinivasa Vittal Katikireddi [13], Kate Tilling [5], Christopher T. Rentsch [15,16], Nicholas J. Timpson [5,6], Nishi Chaturvedi [2] & Claire J. Steves [1,17✉]

The frequency of, and risk factors for, long COVID are unclear among community-based individuals with a history of COVID-19. To elucidate the burden and possible causes of long COVID in the community, we coordinated analyses of survey data from 6907 individuals with self-reported COVID-19 from 10 UK longitudinal study (LS) samples and 1.1 million individuals with COVID-19 diagnostic codes in electronic healthcare records (EHR) collected by spring 2021. Proportions of presumed COVID-19 cases in LS reporting any symptoms for 12+ weeks ranged from 7.8% and 17% (with 1.2 to 4.8% reporting debilitating symptoms). Increasing age, female sex, white ethnicity, poor pre-pandemic general and mental health, overweight/obesity, and asthma were associated with prolonged symptoms in both LS and EHR data, but findings for other factors, such as cardio-metabolic parameters, were inconclusive.

[1] Department of Twin Research and Genetic Epidemiology, School of Life Course Sciences, King's College London, London, UK. [2] MRC Unit for Lifelong Health and Ageing at UCL, University College London, London, UK. [3] Department of Medical Epidemiology and Biostatistics, Karolinska Institutet, Stockholm, Sweden. [4] Bennett Institute for Applied Data Science, Nuffield Department of Primary Care Health Sciences, University of Oxford, Oxford, UK. [5] MRC Integrative Epidemiology Unit at the University of Bristol, Bristol, UK. [6] Population Health Sciences, Bristol Medical School, University of Bristol, Bristol, UK. [7] Institute of Health & Wellbeing, University of Glasgow, Glasgow, UK. [8] Bradford Institute for Health Research, Bradford Teaching Hospitals NHS Foundation Trust, Bradford BD9 6RJ, UK. [9] Centre for Genomic and Experimental Medicine, Institute of Genetics and Cancer, University of Edinburgh, Edinburgh, UK. [10] Division of Psychiatry, University of Edinburgh, Edinburgh, UK. [11] Centre for Longitudinal Studies, UCL Social Research Institute, University College London, London, UK. [12] Department of Epidemiology and Public Health, University College London, London, UK. [13] MRC/CSO Social & Public Health Sciences Unit, University of Glasgow, Glasgow, UK. [14] School of Immunology & Microbial Sciences, King's College London, London, UK. [19] These authors contributed equally: Ellen J. Thompson, Dylan M. Williams, Alex J. Walker. *A list of authors and their affiliations appears at the end of the paper. ✉email: ellen.thompson@kcl.ac.uk; dylan.williams@ucl.ac.uk; claire.j.steves@kcl.ac.uk

S ARS-CoV-2 infection can lead to sustained or recurrent multi-organ symptoms[1–4]. Extended COVID-19 symptomatology over weeks to months has been termed 'long COVID' or post-acute COVID-19 syndrome[5,6]. The UK's National Institute for Health Care and Excellence (NICE) defines COVID-19 symptom duration with three categories: <4 weeks, 4–12 weeks, and >12 weeks), with the latter two categories both considered 'long COVID'[1]. Long COVID prevalence estimates range from 13.3% in highly selected, community-based survey respondents with test-confirmed COVID-19, to at least 71% among those hospitalised by the infection[7–10]. Given the scale of the pandemic, even a low proportion of individuals with long COVID will generate a major burden of enduring illness[11].

Current understanding of long COVID risk factors and its frequency remains poor, impeding mechanistic understanding and intervention and evidence-based service planning. Emerging evidence indicates risk factors for long COVID including demographic characteristics,[5,6,12] comorbidities[5], and immunological response[6]. However, existing studies have often relied on cross-sectional data from small samples. Accurate risk estimates require large generalisable samples with comprehensive measures of pre-pandemic characteristics[13]. UK national primary care records (EHR), covering >95% of the population, afford one data source, but are limited to those seeking care, obtaining a diagnosis of long COVID, and gaining a subsequent diagnostic code. Established population-based longitudinal studies (LS), overcome these limitations by collecting data from participants regardless of healthcare attendance, and benefit from measures of pre-pandemic characteristics. While individual LS are relatively small, combining data from multiple studies yield large sample sizes. Triangulation of findings with equivalent results from EHR can further compensate for different limitations and biases.

To meet clinical and policy needs, we identified individuals with long COVID in: (1) a consortium of population-based LS which captured coordinated repeat questionnaire data on COVID-19 using harmonised measures from the Wellcome Trust's Covid-19 Questionnaire and (2) the OpenSAFELY dataset of primary care records (https://www.opensafely.org/). Here, we report the frequency of long COVID among individuals with suspected and test-confirmed COVID-19 and examined its associations with sociodemographic and pre-pandemic health risk factors.

## Results

**Frequency of Long COVID.** Of 48,901 individuals surveyed in 10 LS samples, 6907 (14.1%) self-reported suspected or confirmed COVID-19 (Table 1 and Supplementary Table 3).

In nine of the LS, respondents with suspected and confirmed COVID-19 self-reported symptom duration according to categories that corresponded to NICE criteria for long COVID (0–4 weeks; 4–12 weeks; 12+ weeks). Reporting of symptoms lasting between 4–12 weeks ranged from 14.5% to 18.1%, and symptoms lasting 12+ weeks ranged from 7.8% to 17%. When restricted to reporting of symptoms that limited day-to-day function, as opposed to reporting of symptoms of any severity, proportions were lower: ranging from 3.0% to 13.7% for 4–12 weeks, and 1.2–4.8% for 12+ weeks (see Table 2).

Frequencies varied considerably within LS when comparing symptom duration reported by self-reported confirmed and suspected cases (see Supplementary Table 4). Among those known to be seropositive for SARS-Cov-2 antibodies or to have received a positive PCR test result, reporting of symptoms for 4–12 weeks ranged from 8.8% to 20%, and 12+ weeks ranged from 11% to 20% (see Supplementary Table 5).

One LS (Born in Bradford) used a different method to ascertain symptom duration, which counted individual symptom reporting over time (recorded retrospectively over several months). We used the same methodology in a second LS (TwinsUK) for comparison. Using this approach, reporting of 4–12 week symptoms ranged between 22.7% and 25.8% and of 12+ week symptoms from 40.9% to 45.6% (Table 2). However, by this method, high proportions of long COVID reporting were also found in those who had not had COVID-19 (21.8% reporting symptoms lasting 4–12 weeks, and 28% reporting symptoms lasting 12+ weeks, Supplementary Table 6). Therefore, data for long COVID ascertained in this way were not taken forward to risk factor analysis due to uncertainty that respondents truly met NICE criteria for a long COVID diagnosis, which require that symptoms are "not explained by an alternative diagnosis".

In the EHR data, among 1,068,680 individuals with any acute COVID-19 diagnostic code, 4189 individuals also had a recorded long COVID code, constituting 0.4% of COVID-19 cases.

**Age and long COVID.** Among LS composed of participants with a range of ages (i.e., age-heterogeneous samples), the risk of symptoms of any severity lasting both 4+ weeks and 12+ weeks increased with higher age were observed across participants ranging from young adulthood to approximately 70 years (Supplementary Figs. 1 and 2). Across age-homogenous LS—where participants within cohort studies were of approximately equal ages ranging from 20 to 63 years—we observe an absolute 3.02% (95% CI: 1.86–4.17) per decade rise in reporting of functionally limiting symptoms lasting 4+ weeks, and a 0.68% (95% CI: −0.15 to 1.51) rise per decade for functionally limiting symptoms lasting 12+ weeks (Fig. 1).

EHR analyses of age and long COVID diagnoses showed an inverted U-shaped association of long COVID risk with age, (Supplementary Fig. 1), with highest risk in those aged 45–54, and 55–69 years. People aged 80 and above had no higher risk than the reference group aged 18–24 years. Among individuals in the EHR sample aged 18–70 years, there was a linear increase of absolute risk of long COVID of 0.12% per decade (95% CI: 0.08–0.17), aligning with LS results (Fig. 1, right panel).

**Sociodemographic factors and long COVID.** Figure 2 shows meta-analysed associations from LS (10 cohorts, n = 6907 cases) between sociodemographic and health factors and each binary long COVID outcome that we analysed: (i) risk of symptoms lasting 4+ weeks versus 0–4 weeks; (ii) risk of symptoms lasting 12+ weeks versus 0–12 weeks. Study-level results are provided in Supplementary Figs. 3–6. Females had higher risk of both long COVID outcomes than males (4+ weeks: OR = 1.49; 95% CI: 1.24–1.79; 12+ weeks: OR = 1.60; 95% CI: 1.23–2.07). No clear evidence was found for individuals of non-white ethnicity having differential risk of symptoms for 4+ weeks compared to individuals of white ethnicity (OR for symptoms lasting 4+ weeks = 0.80; 95%CI: 0.54–1.19). Non-white ethnicity was associated with lower risk of symptoms lasting 12+ weeks specifically (OR = 0.32; 95% CI: 0.22–0.47) after meta-analysis, but study-level findings displayed a high degree of heterogeneity ($I^2$ = 75%, P = 0.001; Supplementary Fig. 5). Across LS, no strong evidence was found for association of IMD with either outcome. Having not attained a degree from higher education was associated with lower risk of symptoms lasting 12+ weeks (OR: 0.73; 95% CI: 0.57–0.94), but not when considering any symptoms lasting 4 weeks or longer (OR: 0.95: 95% CI: 0.80–1.14).

In EHR, females had higher risk of long COVID than males (OR = 1.51; 95% CI:1.41–1.61), while odds were lower in individuals of South Asian (OR = 0.75; 95% CI:0.67–0.84) or Black ethnicity (OR = 0.66; 95% CI:0.52–0.83), relative to white ethnicity (Table 3 and Fig. 2). Individuals living in areas with the

**Table 1 Characteristics of the analytic samples from the longitudinal studies (self-reported COVID-19 cases with data on duration of symptoms).**

| | MCS | ALSPAC G1 | NS | BiB | USoc | BCS70 | TwinsUK | GS | ALSPAC G0 | NCDS |
|---|---|---|---|---|---|---|---|---|---|---|
| Sample size | 1055 | 668 | 848 | 110 | 1033 | 889 | 806 | 335 | 446 | 709 |
| Age, mean years (SD) | 19.9 (0.3) | 28.4 (0.5) | 31.0 (0.3) | 40.7 (5.9) | 48.5 (14.8) | 51[a] | 52.7 (15.8) | 55.9 (10.6) | 58.3 (4.4) | 63[a] |
| Female sex, N (%) | 652 (61.8) | 426 (63.8) | 539 (64.6) | 106 (96.4) | 675 (65.3) | 507 (57.0) | 709 (88) | 215 (64.2) | 303 (67.9) | 389 (54.9) |
| *Ethnicity, N (%)* | | | | | | | | | | |
| White | 862 (81.7) | 638 (95.5) | 574 (67.7) | 49 (44.5) | 879 (85.1) | 747 (84.0) | 776 (96.3) | 330 (98.5) | 439 (98.4) | 652 (92.0) |
| Non-white minority ethnic group | 192 (18.2) | 30 (4.5) | 254 (30.0) | 56 (50.9) | 136 (13.2) | 27 (3.0) | 30 (3.7) | 5 (1.5) | 6 (1.3) | 19 (2.7) |
| Missing | 1 (0.1) | 0 | 20 (2.4) | 5 (4.6) | 18 (1.7) | 115 (12.9) | 1 (0.1) | 0 | 1 (0.2) | 35 (5.4) |
| *Education, N (%)* | | | | | | | | | | |
| Degree | 494 (46.8) | 338 (50.6) | 396 (49.7) | 11 (10) | 500 (48.4) | 377 (42.4) | 402 (49.9) | 163 (48.7) | 106 (23.8) | 284 (40.1) |
| No degree | 502 (47.6) | 149 (22.3) | 358 (42.2) | 82 (74.5) | 429 (41.5) | 444 (49.9) | 224 (27.8) | 165 (49.3) | 307 (68.8) | 415 (58.5) |
| Missing | 59 (5.6) | 181 (27.1) | 94 (11.1) | 17 (15.5) | 104 (10.1) | 68 (7.7) | 180 (22.3) | 7 (2.1) | 33 (7.4) | 10 (1.4) |
| *Occupational class, N (%)* | | | | | | | | | | |
| Managerial, Admin, Professional | - | 120 (18.0) | - | 26 (23.6) | 402 (38.9) | - | - | 177 (52.8) | 57 (12.8) | - |
| Intermediate | - | 280 (41.9) | - | 36 (32.7) | 171 (16.6) | - | - | 60 (17.9) | 130 (29.1) | - |
| Manual/Routine | - | 171 (25.6) | - | 21 (19.1) | 220 (21.3) | - | - | 37 (11.0) | 190 (42.6) | - |
| Not in employment | - | 2 (0.3) | - | - | 212 (20.5) | - | - | - | 5 (1.1) | - |
| Missing | - | 95 (14.2) | - | 27 (24.5) | 28 (2.7) | - | - | 61 (18.2) | 64 (14.3) | - |
| *Country, N (%)* | | | | | | | | | | |
| England | 746 (70.7) | 668 (100) | 828 (97.6) | 110 (100) | 866 (83.8) | 770 (86.6) | 747 (92.7) | 4 (1.2) | 446 (100) | 613 (86.5) |
| Scotland | 93 (8.8) | - | 5 (0.6) | - | 62 (6.0) | 57 (6.4) | 26 (3.2) | 331 (98.8) | - | 45 (6.4) |
| Wales | 136 (12.9) | - | 9 (1.1) | - | 69 (6.7) | 44 (5.0) | 24 (3.0) | - | - | 38 (5.4) |
| Northern Ireland | 75 (7.1) | - | 1 (0.1) | - | 36 (3.5) | 0 | 1 (0.1) | - | - | 2 (0.3) |
| Missing/Other | 5 (0.5) | - | 5 (0.6) | - | 0 (0) | 18 (2.0) | 8 (1.0) | - | - | 11 (1.6) |
| Hospitalised with COVID-19, N (%) | 8 (0.8) | - | 23 (2.7) | - | 21 (2.0) | 40 (4.5) | 27 (3.3) | 3 (4.5) | - | 37 (5.2) |

Studies are ordered left to right from youngest to oldest mean age.
*MCS* Millennium Cohort Study, *ALSPAC G1* Children of the Avon Longitudinal Study of Parents and Children, *NS* next steps, *BiB* born in Bradford, *USoc* understanding society, *BCS70* 1970 British Cohort Study, *TwinsUK* UK Adult Twin Registry, *GS* Generation Scotland, ALSPAC G0 parents of ALSPAC, *NCDS* National Child Development Study.
[a]Age-homogeneous cohorts.

**Table 2 Symptoms duration among self-reported COVID-19 cases in the longitudinal studies.**

| Study | COVID-19 cases with symptom duration data | Mean age | Duration of symptoms, N (%) | | |
|---|---|---|---|---|---|
| | | | Acute (0–4 weeks) | Ongoing symptomatic COVID-19 (4–12 weeks) | Post COVID-19 syndrome (12+ weeks) |
| *Studies ascertaining long COVID of any severity* | | | | | |
| ALSPAC G1 | 668 | 28.4 | 519 (77.7) | 97 (14.5) | 52 (7.8) |
| USoc | 1033 | 48.5 | 742 (71.8) | 182 (17.6) | 109 (10.6) |
| TwinsUK | 806 | 52.7 | 579 (71.8) | 146 (18.1) | 81 (10) |
| GS | 335 | 55.9 | 224 (66.9) | 54 (16.1) | 57 (17.0) |
| ALSPAC G0 | 446 | 58.3 | 302 (67.7) | 68 (15.2) | 76 (17.0) |
| *Studies ascertaining severe long COVID only*[a] | | | | | |
| MCS | 1055 | 19.9 | 1010 (95.7) | 32 (3.0) | 13 (1.2) |
| Next Steps | 848 | 31.0 | 773 (91.2) | 51 (6.0) | 24 (2.8) |
| BCS70 | 889 | 51.0 | 757 (85.2) | 84 (9.5) | 48 (5.4) |
| NCDS | 709 | 63.0 | 578 (81.5) | 97 (13.7) | 34 (4.8) |
| *Studies ascertaining long COVID by monthly symptom reporting*[b] | | | | | |
| BiB | 110 | 40.7 | 40 (36.4) | 25 (22.7) | 45 (40.9) |
| TwinsUK | 953 | 54 | 272 (28.5) | 246 (25.8) | 435 (45.6) |

Studies are ordered from youngest to oldest mean age within categories of method of long COVID ascertainment.
*ALSPAC* Avon Longitudinal Study of Parents and Children (Generations 0 and 1), *BCS70* 1970 British Cohort Study, *BiB* born in Bradford, *GS* generation Scotland, *MCS* Millennium Cohort Study, *NCDS* 1958 National Child Development Study, *NS* next steps, *USoc* Understanding Society.
[a]Questionnaires in these four cohorts asked respondents to report duration for which COVID-19 symptoms impeded normal function, rather than simply the duration of any symptoms (however mild) as in other studies. Hence proportions reporting long COVID in them are expected to be lower when compared to other cohorts with similar characteristics.
[b]Based on symptom-counting approach over months, rather than self-reported duration of symptoms as in all other cohorts, which yields higher proportions of individuals being designated long COVID categories.

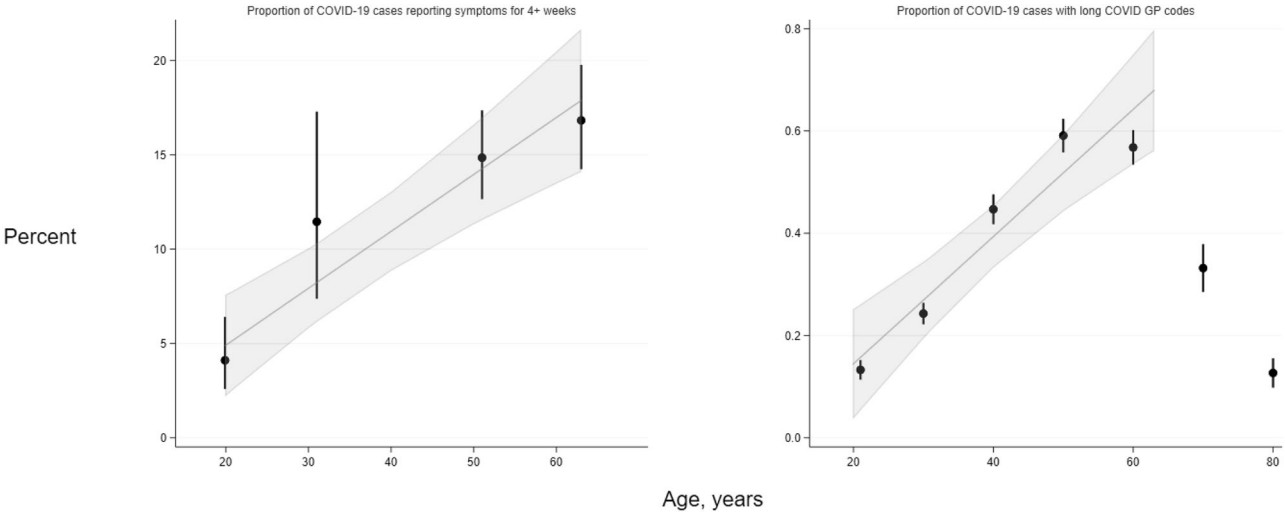

**Fig. 1 Trends in long COVID frequency among COVID-19 cases by age, in four age-homogeneous LS (left) and EHRs (right).** Left—in four longitudinal studies (MCS N = 1055; NS N = 848; BCS70 N = 889; NCDS N = 709) where participants are of near-identical ages, proportions reporting symptom length of four or more weeks in COVID-19 cases were ascertained from questionnaire responses. Right–in OpenSAFELY (N = 4189), proportions represent individuals within 10-year age categories (with estimates grouped at the mid-point of each category) who have long COVID codes in GP records, hence the proportions are substantially lower than in the corresponding cohort data. Data are presented as percentages and 95% confidence intervals (CIs) as appropriate. Trend lines and 95% CIs shading represent absolute differences in long COVID frequencies with increasing age, estimated by linear meta-regression of data from the four cohorts and from 18- to 70-year-olds in OpenSAFELY (data from older individuals were not modelled; refer to results text for further explanation).

least deprivation had higher odds of a long COVID code compared to those in the most deprived IMD quintile (Fig. 2).

*Health factors and long COVID.* In LS, those with poor or fair pre-pandemic self-reported general health had greater risk of both long COVID outcomes (4+ weeks: OR = 1.62; 95%CI: 1.25–2.09; 12+ weeks: OR = 1.66; 95%CI: 1.14–2.40). Greater pre-pandemic psychological distress was also associated with higher risk of both long COVID outcomes (4+ weeks: OR = 1.45; 95% CI: 1.16–1.82; 12+ weeks: OR = 1.58; 95% CI: 1.15–2.17). No strong evidence was observed for a linear association of BMI with either outcome, while overweight/obesity was associated

with increased odds of symptoms lasting for 4+ weeks (OR = 1.24; 95% CI: 1.01–1.53) but not with symptoms lasting 12+ weeks specifically (OR 0.95, 95% CI: 0.70–1.28). There was no strong evidence of associations of either long COVID outcome with diabetes, hypertension, or high cholesterol with either outcome, although modest point estimates were on the side of higher long COVID risk in several instances (Supplementary Figs. 4 and 6). Asthma was the only specific medical condition associated with increased odds of having symptoms for 4+ weeks (OR = 1.31; 95% CI: 1.06–1.62), although the association with symptoms lasting 12+ weeks specifically was closer to the null (OR = 1.14; 95% CI: 0.83–1.57).

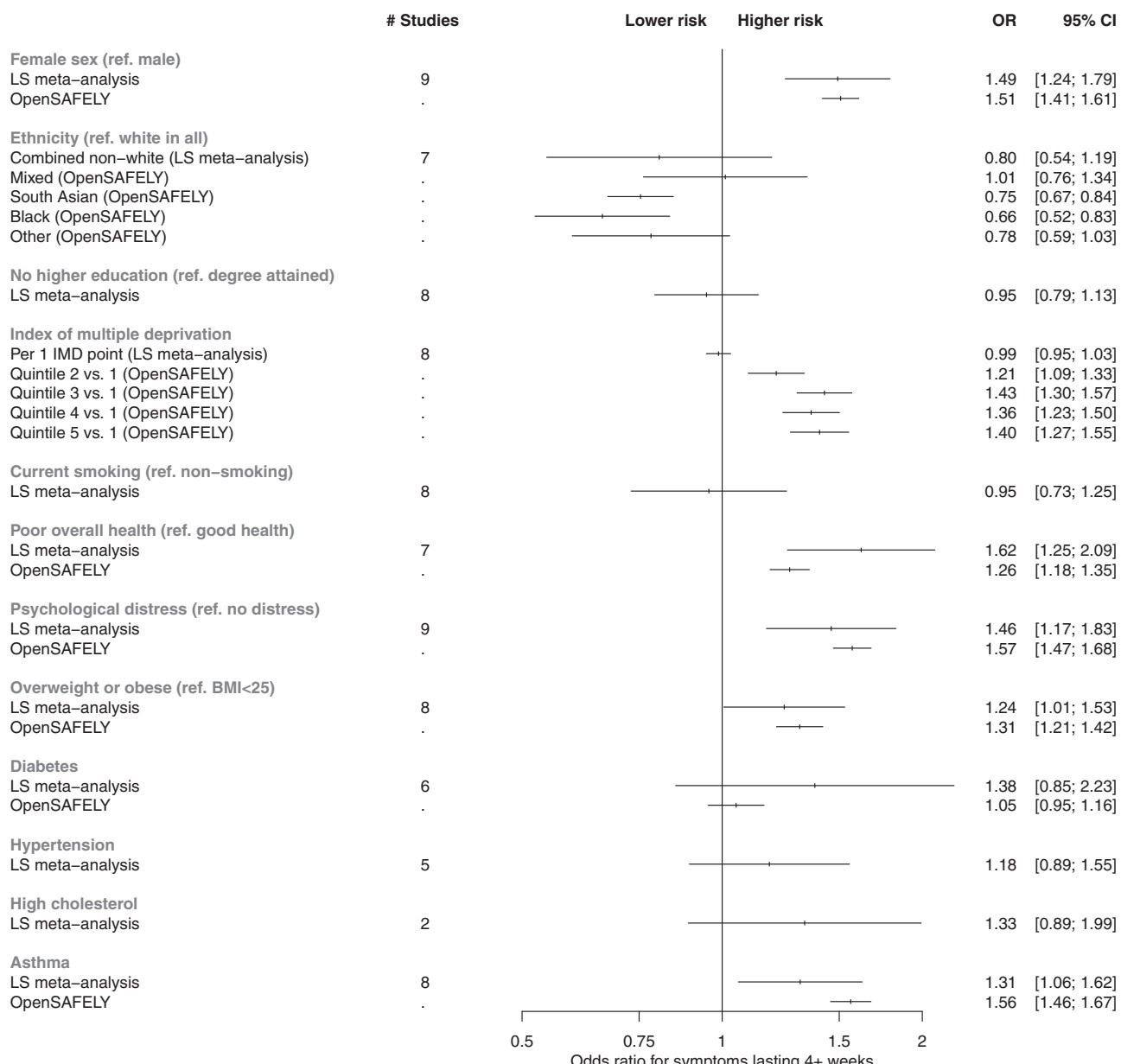

**Fig. 2 Risk factors associated with long COVID from meta-analyses of LS findings alongside corresponding analyses from EHRs.** The reference category for 'Diabetes', 'Hypertension', 'High Cholesterol', and 'Asthma' is the absence of condition. All associations were adjusted for age and sex, except where redundant. In all instances where it was possible to derive results from both meta-analyses of longitudinal studies ($N$ up to = 6907) and analysis of EHRs ($N$ up to = 4189), the corresponding results are plotted side-by-side for comparison. Estimates from fixed effects meta-analyses of longitudinal study data and EHR analyses are presented as odds ratios (OR) and 95% confidence intervals (CIs). The outcome used for longitudinal study analyses presented here was symptoms lasting for 4+ weeks, and the outcome in EHRs was any reporting of a long COVID read code in GP records (regardless of duration of symptoms). Full study-level results, heterogeneity statistics and random-effect estimates for the longitudinal study meta-analyses are presented in Supplementary Figs. 3 and 4. The equivalent meta-analyses of longitudinal study data where symptom duration of 12+ weeks was instead used as the outcome are depicted in Supplementary Figs. 5 and 6. Index of multiple deprivation quintile 1 represents individuals from the most deprived area, and quintile 5 represents individuals from the least deprived area. 'Poor overall health' represents the self-rated health exposure in the LS meta-analysis, and comorbidities in OpenSAFELY. The outcome 'Overweight and obesity' represents combined BMI categories over 25 in the LS, and solely individuals with BMI 30–34.9 in OpenSAFELY.

In EHR, increased odds of having a long COVID code was seen in individuals with pre-existing comorbidities (OR = 1.26; 95% CI:1.18–1.35) and psychiatric conditions (OR = 1.57; 95% CI:1.47–1.68). An increased risk was observed in individuals with a pre-pandemic diagnosis of asthma (OR = 1.56; 95% CI: 1.46–1.67) and overweight and obesity (OR = 1.31, 95% CI: 1.21–1.42). No increase in risk was observed for diabetes (OR = 1.05, 95% CI: 0.95–1.16).

*Sensitivity analyses.* In LS, when using inverse-probability weighting (IPW) to account for the possibility that long COVID associations were influenced by index event bias from risk factor associations with risk of COVID-19, all identified associations persisted and, in some instances, associations increased slightly in magnitude (Supplementary Figs. 7–10). Notably hypercholesterolaemia was associated with both long COVID outcomes in the LS meta-analyses weighted for probability of reporting COVID-19.

**Table 3 Characteristics of individuals reported to have had COVID-19 and long COVID by general practitioners in OpenSAFELY.**

|  | Acute COVID-19 | Long COVID | Long COVID rate per 100,000 cases | Proportion of long COVID cases in category (%) |
|---|---|---|---|---|
| Sample size | 1,064,491 | 4189 | 392 | 0.4 |
| *Age, years* |  |  |  |  |
| 18–24 | 137,997 | 184 | 133.2 | 4.4 |
| 25–34 | 211,479 | 515 | 242.9 | 12.3 |
| 35–44 | 199,750 | 897 | 447.1 | 21.4 |
| 45–54 | 208,351 | 1238 | 590.7 | 29.6 |
| 55–69 | 190,616 | 1088 | 567.5 | 26 |
| 70–79 | 57,886 | 193 | 332.3 | 4.6 |
| 80+ | 58,412 | 74 | 126.5 | 1.8 |
| *Sex* |  |  |  |  |
| Female | 582,220 | 2678 | 457.9 | 63.9 |
| Male | 482,271 | 1511 | 312.3 | 36.1 |
| *Ethnicity* |  |  |  |  |
| White | 635,414 | 2647 | 414.9 | 63.2 |
| Mixed | 12,498 | 49 | 390.5 | 1.2 |
| South Asian | 111,026 | 340 | 305.3 | 8.1 |
| Black | 25,886 | 73 | 281.2 | 1.7 |
| Other | 16,521 | 53 | 319.8 | 1.3 |
| *IMD quantile* |  |  |  |  |
| Missing | 22,104 | 75 | 338.2 | 1.8 |
| 1 | 255,431 | 787 | 307.2 | 18.8 |
| 2 | 226,760 | 850 | 373.4 | 20.3 |
| 3 | 208,684 | 932 | 444.6 | 22.2 |
| 4 | 188,224 | 814 | 430.6 | 19.4 |
| 5 | 163,288 | 731 | 445.7 | 17.5 |
| *BMI category* |  |  |  |  |
| Not obese | 800,439 | 2694 | 335.4 | 64.3 |
| Obese I (30–34.9) | 151,782 | 787 | 515.8 | 18.8 |
| Obese II (35–39.9) | 67,470 | 411 | 605.5 | 9.8 |
| Obese III (40+) | 44,800 | 297 | 658.6 | 7.1 |
| *Health conditions* |  |  |  |  |
| 0 | 661,200 | 2336 | 352.1 | 55.8 |
| 1 | 291,106 | 1335 | 456.5 | 31.9 |
| 2 or more | 112,185 | 518 | 459.6 | 12.4 |
| *Mental health disorder(s)* |  |  |  |  |
| 0 | 835,361 | 2772 | 330.7 | 66.2 |
| 1 or more | 229,130 | 1417 | 614.6 | 33.8 |
| *Asthma* |  |  |  |  |
| No | 872,030 | 3129 | 357.5 | 74.7 |
| Yes | 192,461 | 1060 | 547.7 | 25.3 |
| *Diabetes* |  |  |  |  |
| No | 951,029 | 3686 | 386.1 | 88.0 |
| Yes | 113,462 | 503 | 441.4 | 12.0 |

*BMI* body mass index, *IMD* index of multiple deprivation (quantile 1 representing most deprived, and 5 representing least deprived).

In sensitivity analyses from three LS that had a subset of COVID-19 cases confirmed by a positive PCR test and/or serology testing, meta-analysed associations of sociodemographic factors with outcomes were broadly similar, except that there were pronounced associations of lower educational attainment with lower risk of both outcomes (Supplementary Figs. 11 and 13). This disparity might have arisen due to bias from more educated participants being more likely to seek testing and being selected into the case samples. In meta-analyses of health factors with the outcomes in these subgroups, several associations with both outcomes were more modest and included the null when compared to findings from the full sample meta-analyses (Supplementary Figs. 12 and 14). These included results for asthma and pre-pandemic overall health and psychological distress. However, we note that the precision of these estimates was limited.

## Discussion

In parallel analyses of 10 population-based longitudinal studies and 1.2 million primary care EHRs, we observed varying proportions of adults with COVID-19 who had long COVID depending on the age of study members and whether the symptoms limited day to day functioning. While just 0.3% of COVID-19 cases had long COVID codes in primary care, up to 17% of adult COVID-19 cases in midlife reported symptoms attributed to COVID-19 for more than 12 weeks in longitudinal studies. Clear associations between long COVID risk and sociodemographic characteristics (older age, female sex, white ethnicity) and antecedent health factors (poor mental and general health, asthma) were also established.

Recent reports of the frequency of long COVID vary, with the Real-time Assessment of Community Transmission (REACT)-2 study reporting 14.8% of COVID-19 cases with 3 or more

symptoms persisting for 12+ weeks, and 11.5% of COVID-19 cases with 3 or more enduring symptoms affecting their daily lives[14]. These estimates are significantly higher than our estimates of functionally limiting symptoms lasting 12+ weeks (1.2–4.8%, according to age). However, as detailed above, this definition diverges from the NICE definition which requires symptoms not to be attributable to an alternative cause. Using a symptom counting approach in our study, we found that the proportions of symptoms lasting for 4+ weeks and 12+ weeks were consistent with other population-based studies[14–16]. However, high rates of symptom reporting were also found in those without COVID-19, thus estimates using this approach should be treated with caution. Notable discordance in these proportions would yield very different prevalence estimates for the number of people in the UK population that might require care for long COVID, with REACT-2 estimating 5.8% of the English population would self-report any degree of long COVID (i.e. not necessarily reporting debilitating symptoms) to early February 2021, whereas the Office National Statistics (ONS) estimated 1.5% of the UK population would self-report any degree of prolonged symptoms (and ~0.3% would self-report symptoms to be limiting day-to-day activities a lot) as of June 2021. Several reasons could explain disparities in observed proportions with long COVID, including estimates for England vs. the UK as a whole, questionnaire wording and timing of field work, examining symptoms regarded as debilitating vs. symptoms of any severity, basing estimates on test-validated versus self-reported COVID-19 cases, and representativeness (REACT-2's response rates being 26–29%; ONS reporting 51% for its May 2021 survey; and in the most recent LS surveys, response rates for studies that reported functionally limiting symptoms ranged from 33% to 58.5%).

The lower reporting of long COVID in primary care compared to our LS data and other studies suggest that only a minority of people with long COVID seek care and/or subsequently receive a code. Diagnostic codes for long COVID have only recently been instituted and uptake by primary care practitioners has not been uniform[17]. The analyses here are based on practices that use TPP SystmOne software and is therefore limited to England, and we note that these practices had a 2- to 3-fold lower rate of long COVID recording than those that use EMIS software[17].

Despite definition differences in primary care versus LS, several risk factor associations were consistent between various LS and in EHR. In both LS and EHR, long COVID reporting by any definition increased with age. Unlike risk of severe COVID-19, this appeared to be linear (not exponential) across most adult age groups. In individuals aged over 70 we observed a sharp decline in long COVID risk in most LS and the EHR data. This decline in older age has been observed in other studies[5,18,19], and may be spurious due to selective competing risk of mortality, non-response bias, lower symptom reporting in older adults, misattribution of long COVID to other illness, or a combination of these factors. The findings that the odds of long COVID was 50% higher in women than men is consistent with reports from most[5,18,20–23] but not all previous studies[5,19]. We found some evidence of higher long COVID reporting among individuals of white ethnicity and of higher educational attainment, which was unexpected given the common associations of these characteristics with lower morbidity more generally. While we found no strong evidence for a relationship between area-level socioeconomic status in LS, in primary care EHR there was also an apparent gradient of higher risk in individuals from the least deprived areas. These associations could reflect unmet need in medical care for those who live in socioeconomically deprived areas or circumstances. However, these results contrast with two studies reporting null findings for ethnicity and socioeconomic status in relation to long COVID from other countries[19,24], and the ONS and REACT-2 surveys reported similar associations for ethnicity, but opposite associations in long COVID reporting by deprivation scales, to that which we observed in EHR[14].

A greater risk of long COVID related to adverse prior mental health, has been reported elsewhere[24], but pre-pandemic general health has not previously been highlighted as a risk factor[18,21,25]. Excess risk of long COVID in association with asthma across cohorts and primary care records combats previous conflicting and limited findings[5,6,18,24], and supports a focus on asthma as a high-risk condition, for example by investigating whether immune processes involved in asthma or respiratory complications influence long COVID development. Findings for overweight/obesity were suggestive of an increased risk, again helping to resolve some previous uncertainty[5,18,21,26]. No other cardio-metabolic risk factors were clearly associated, consistent with past studies[5,21,24,26].

A major strength of this research was the coordinated investigation of long COVID in multiple LS and EHR, each with differing bias, study designs, target populations, and selection and attrition processes. Consistent findings emerging from these sources add reliability. We used population-based resources to increase the representativeness of findings to long COVID in the community. Unlike newly established studies which have collected exposure data during the pandemic, the long-running data collections in both LS and EHR allowed us to study *prospective* associations of risk factors with long COVID, meaning results will not have arisen from reverse causation, nor will exposure definitions have been influenced by recall bias. Rich antecedent data also allowed us to run a range of sensitivity analyses to re-weight our results for non-response (reducing the bias from selection into samples). We also flag important limitations, principally that our data are observational, and we cannot draw causal conclusions on the role of risk factors in long COVID development, and that whilst we attempted to address both selection into our samples from study attrition and selecting upon COVID-19 case status (which can induce index event bias)[27], there remains the possibility that potential bias has influenced association estimates. However, we attempt to reduce the likelihood of index event bias in the LS sample though the use of inverse probability weighting on the probability of getting COVID-19. Finally, not all studies had test confirmation of COVID-19 status, and some individuals may have misattributed persistent symptoms to other conditions. From past analyses to establish case definitions in two of the samples (ALSPAC G0 and G1), 25.8% and 32.2% of self-reported cases could be verified against PCR results from linked national testing and/or serology, respectively. Though this implies that there may be many self-reported COVID-19 instances in the samples prone to misclassification, there are issues with test confirmation that mean true misclassification may be much lower (e.g., limited surveillance with PCR testing, imperfect sensitivity of both PCR and antibody tests, and waning antibody titres leading to seroreversion over time). The impact of bias from misclassification on the risk factor associations with long COVID is unclear. Sensitivity analysis of those with positive PCR/antibody data showed some inconsistencies in directions of associations. However, these results should be interpreted cautiously due to the small sample sizes included, and further collections of test data on large-scale LS will be required to augment the number of confirmed cases for similar analyses in future.

**Implications**. The stark variability in proportions of COVID-19 cases with persistent symptoms is clear from our comparison of methods of ascertaining long COVID. Representative population-based studies will need to provide ongoing estimates across the spectrum of functional limitation to help plan appropriate provision of healthcare. Our data suggest that revisions of diagnostic criteria within primary care may be appropriate, particularly for demographic groups which are less in touch with healthcare

services. Although causal inferences cannot be drawn from these data, our findings justify further investigations into the role of sex difference, age related change, and/or immunity and respiratory health in development of long COVID. Older working individuals, with high levels of comorbidity, may particularly require support.

## Methods

**Design**. The UK National Core Studies—Longitudinal Health and Wellbeing programme (https://www.ucl.ac.uk/covid-19-longitudinal-health-wellbeing/) combines data from multiple UK population-based LS and electronic health records (EHR) to answer pandemic-relevant questions. In this analysis we pooled results from parallel analyses within individual LS, then compared with population-based findings from EHR capturing individuals who actively sought healthcare.

## Sample

*LS.* Data were drawn from 10 UK LS that had conducted surveys before and during the COVID-19 pandemic comprising five age-homogenous cohorts: the Millennium Cohort Study (MCS)[28]; the Avon Longitudinal Study of Parents and Children (ALSPAC (generation 1, "G1"))[29]; Next Steps (NS)[30]; the 1970 British Cohort Study (BCS)[31]; and the National Child Development Study (NCDS)[32], and five age-heterogeneous samples were included: the Born in Bradford study (BIB)[33]; Understanding Society (USOC)[34]; Generation Scotland: the Scottish Family Health Study (GS)[35]; the parents of the ALSPAC-G1 cohort, whom we refer to as ALSPAC-G0[36]; and the UK Adult Twin Registry (TwinsUK)[37]. Study details and references are shown in Supplementary Table 1. Minimum inclusion criteria were pre-pandemic health measures, age, sex, ethnicity plus self-reported COVID-19, and self-reported duration of COVID-19 symptoms. Ethics statements presented in Supplementary Table 2.

*Electronic health records (EHR).* Working on behalf of NHS England, we conducted a population-based cohort study to measure long COVID recording in EHR data from primary care practices using TPP SystmOne software, linked to Secondary Uses Service (SUS) data (containing hospital records) through OpenSAFELY (https://www.opensafely.org/). This is a data analysis platform developed on behalf of NHS England during the COVID-19 pandemic to allow near real-time analysis of pseudonymised primary care records within the EHR vendor's highly secure data environment to protect patient privacy. Details on Information Governance for the OpenSAFELY platform can be found in the Supplementary Note 1. From a population of all people alive and registered with a general practice on 1 December 2020, we selected all patients who had evidence of a COVID-19-related code, either: positive SARS-CoV-2 testing, being hospitalised with an associated COVID diagnostic code, or having a recorded diagnostic code for COVID in primary care.

## Measures

*Outcomes: COVID-19 and long COVID definitions.* LS: COVID-19 cases were defined by self-report, including testing confirmation and healthcare professional diagnosis (see Supplementary Data 1 for full details of the questions and coding used within each study). Long COVID was defined as per NICE categories using self-reported symptom duration[1]. Based on these categories, we defined two primary outcomes: (i) symptoms lasting 4+ weeks (symptoms lasting 0–4 weeks as reference) and (ii) symptoms lasting 12+ weeks (symptoms lasting 0–12 weeks as reference). Some studies recorded duration of symptoms of any severity, whereas others referred only to symptoms which impacted daily function (Table 2). In addition, two studies derived alternate estimates of long COVID based on individual symptom counts lasting more than 4 or 12 weeks over at least six months (BiB, TwinsUK) (Supplementary Note 2). All data used to derive these outcomes were collected between April and November 2020.

EHR: Any record of long COVID in the primary care record was coded as a binary variable. This was defined using a list of 15 UK SNOMED codes, categorised as diagnostic (2 codes), referral[3] and assessment[10] codes. SNOMED is an international structured clinical coding system for use in EHR[38]. These clinical codes were designed based on guidance issued on long COVID by the NICE[1]. The outcome was measured between the study start date (1 February 2020) and the end date (9 May 2021).

## Exposures

*Sociodemographic factors.* All studies included age, sex, ethnicity (white or non-white minority ethnic group, where available) and Index of Multiple Deprivation (IMD; divided into quintiles with 1 representing the most deprived and 5 representing the least deprived). Area-level SES was measured using the IMD 2019, a composite of different domains including area-level income, employment, education access and crime, for the postcode where a participant lived at the time of sample collection[39]. LS included additional measures of socioeconomic position: education (degree, no degree), and occupational class of own current/recent employment (Supplementary Data1). EHR also included geographic region[40].

**Mental health**. LS: Pre-pandemic measures using validated continuous scales of anxiety and depression symptoms dichotomised using established cut-offs to indicate distress (see Supplementary Data 1).

EHR: Evidence of a pre-existing mental health condition was defined using prior codes for one of: psychosis; schizophrenia; bipolar disorder; or depression.

**Self-rated general health**. LS: Pre-pandemic self-rating on a 5-point scale dichotomised to compare excellent-good health (categories 1–3) with fair-poor health (categories 4 and 5).

**Overweight and obesity**. LS: Body mass index (BMI; kg/m$^2$) obtained prior to the pandemic, coded to compare a BMI between 0 and 24.9 (having underweight/ normal weight) against a BMI of ≥25 (overweight/obesity).

EHR: Categorised as having or not having obesity using the most recent BMI measurement, with those having obesity further classified into having Obese I (BMI 30–34.9), Obese II (BMI 35–39.9), or Obese III (BMI 40+). A BMI of >25 was used in LS as the percentage of those in the obese category (i.e., BMI > 30) was relatively small, e.g., 8.9% for TwinsUK, whereas EHR obesity codes were used as these are more reliable and valid indicators of having obesity in general practice.

**Health conditions**. LS: Pre-pandemic self-report of asthma, diabetes, hypertension, and high cholesterol status.

EHR: A previous code 6 months to 5 years before March 2020 for one or more of: diabetes; cancer; haematological cancer; asthma; chronic respiratory disease; chronic cardiac disease; chronic liver disease; stroke or dementia; other neurological condition; organ transplant; dysplasia; rheumatoid arthritis, systemic lupus erythematosus or psoriasis; or other immunosuppressive conditions. Those with no relevant code for a condition were assumed not to have that condition. Number of conditions were categorised into "0", "1", and "2 or more".

**Health behaviours**. LS: Current smoking status (dichotomised into "0" = no, "1" = yes).

**Statistical analysis: LS**. Main analyses were conducted in studies with a direct self-reported measure of COVID-19 symptom length. Associations between each factor and both long COVID outcomes (symptoms for 4+ weeks and symptoms for 12+ weeks) were assessed in separate logistic regression models within each study. We adjust for a minimal set of covariates across all studies, where relevant: age (adjusted as a continuous variable when being considered a covariate), sex, and ethnicity. We report odds ratios (ORs) and 95% confidence intervals (CIs). To synthesise association magnitudes across studies, fixed-effect meta-analysis with restricted maximum likelihood was carried out and repeated with random-effects modelling for comparison. The $I^2$ statistic was used to report heterogeneity between estimates. Meta-analyses were conducted using the metafor package[41] for R version 4).

Due to the different age structures of the LS, examination of the direct relationship of age with long COVID risk was treated distinctly from other risk factors, and we modelled the relationship in two ways. First, in age-heterogeneous samples we compared long COVID risk within pre-defined age categories, comparing 45–69 and 70+ to 18–44 in three cohorts (USOC, TwinsUK and GS), and 55–59 and 60–76 to 45–54 in one cohort (ALSPAC G0). Second, in a subset of LS birth cohorts with participants of near-identical ages and who were issued fully harmonised long COVID questionnaires (MCS, NS, BCS70 and NCDS), we analysed the trend in absolute risk of long COVID with increasing age between studies using meta-regression.

Attrition and survey design were addressed by weighting estimates to be representative of their target population in each LS (weights were not available for BiB and TwinsUK).

**Sensitivity analyses**. To mitigate index event bias[27], IPW were derived for risk of COVID-19. These were derived in each LS separately but following a common approach used previously (see Supplementary Note 3 for detail)[42]. Derived weights were then applied in all analysis models as a sensitivity check.

For studies in which we were able to verify SARS-CoV-2 infection (TwinsUK and ALSPAC-G0 and -G1), analyses were repeated on the sub-sample of those who had positive polymerase chain reaction (PCR) obtained through linkage to testing data and/or lateral flow antibody testing (ALSPAC) and enzyme-linked immunosorbent assay (ELISA) (TwinsUK)[43] results confirming viral exposure. These results are presented in Supplementary Figs. 11–14.

**Statistical analysis: EHR**. We conducted logistic regression to assess whether GP-recorded long COVID was associated with each sociodemographic or pre-pandemic health characteristic. We adjusted for the same set of confounders as used in the LS analyses: age (as categorical variable), sex, ethnicity.

In further analyses of age as a risk factor for long COVID in the EHR data, we assigned individuals within 10-year categories an age at the midpoint of each group, then assessed the trend in long COVID frequency with age using linear and non-linear meta-regression.

**Reporting summary**. Further information on research design is available in the Nature Research Reporting Summary linked to this article.

## Data availability

Data access for NCDS (SN 6137), BCS70 (SN 8547), Next Steps (SN 5545), MCS (SN 8682) and all four COVID-19 surveys (SN 8658) can be obtained through the UK Data Service. ALSPAC data is available to researchers through an online proposal system. Information regarding access can be found on the ALSPAC website (http://www.bristol.ac.uk/media-library/sites/alspac/documents/researchers/data-access/ALSPAC_Access_Policy.pdf). Data from the various Born in Braford family studies are available to researchers; see the study website for information on how to access data (https://borninbradford.nhs.uk/research/how-to-access-data/). Generation Scotland data are available through the UK Data Service (SN 6614 and SN 8644). Access to data is approved by the Generation Scotland Access Committee. See https://www.ed.ac.uk/generation-scotland/for-researchers/access or email access@generationscotland.org for further details. TwinsUK data are available on request from the TwinsUK Resource Executive Committee (TREC). Access to TwinsUK data can be obtained via a standard application procedure. Data requests should be submitted via https://twinsuk.ac.uk/resources-for-researchers/access-our-data/.

## Code availability

Analysis code for the meta-analyses and forest plotting of long COVID risk factors from 10 LS samples can be accessed on https://github.com/dylwil/longCOVIDrisk. All code for the OpenSAFELY platform for data management, analysis and secure code execution is shared for review and re-use under open licenses at https://github.com/opensafely. All codelists (describing the definition of the conditions) and the code for data management and analysis is shared for scientific review and re-use under open licenses on GitHub https://github.com/opensafely/long-covid-historical-health, with the code archived on Zenodo (https://doi.org/10.5281/zenodo.6361864).

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

## Acknowledgements

This work was supported by the National Core Studies, an initiative funded by UKRI, NIHR, and the Health and Safety Executive. The COVID-19 Longitudinal Health and Wellbeing National Core Study was funded by the Medical Research Council (MC_PC_20030). Related funding was also provided by the NIHR (CONVALESCENCE grant COV-LT-0009). The contributing studies have been made possible because of the tireless dedication, commitment and enthusiasm of the many people who have taken part. We would like to thank the participants and the numerous team members involved in the studies including interviewers, technicians, researchers, administrators, managers, health professionals and volunteers. We are additionally grateful to our funders for their financial input and support in making this research happen. Data gathered from questionnaires was provided by Wellcome Longitudinal Population Study (LPS) COVID-19 Steering Group and Secretariat (221574/Z/20/Z). Understanding Society is an initiative funded by the Economic and Social Research Council and various Government Departments, with scientific leadership by the Institute for Social and Economic Research, University of Essex, and survey delivery by NatCen Social Research and Kantar Public. The Understanding Society COVID-19 study is funded by the Economic and Social Research Council (ES/K005146/1) and the Health Foundation (2076161). The research data are distributed by the UK Data Service. The Millennium Cohort Study, Next Steps, British Cohort Study 1970 and National Child Development Study 1958 are supported by the Centre for Longitudinal Studies, Resource Centre 2015-20 grant (ES/M001660/1) and a host of other co-funders. The COVID-19 data collections in these five cohorts were funded by the UKRI grant Understanding the economic, social and health impacts of COVID-19 using lifetime data: evidence from 5 nationally representative UK cohorts (ES/V012789/1). The UK Medical Research Council and Wellcome (Grant Ref: 217065/Z/19/Z) and the University of Bristol provide core support for ALSPAC. A comprehensive list of grants funding is available on the ALSPAC website (http://www.bristol.ac.uk/alspac/external/documents/grant-acknowledgements.pdf). We are extremely grateful to all the families who took part in this study, the midwives for their help in recruiting them, and the whole ALSPAC team, which includes interviewers, computer and laboratory technicians, clerical workers, research scientists, volunteers, managers, receptionists, and nurses. TwinsUK receives funding from the Wellcome Trust (WT212904/Z/18/Z), the National Institute for Health Research (NIHR) Biomedical Research Centre based at Guy's and St Thomas' NHS Foundation Trust and King's College London. The TwinsUK COVID-19 personal experience study was funded by the King's Together Rapid COVID-19 Call award, under the projects original title 'Keeping together through coronavirus: The physical and mental health implications of self-isolation due to the Covid-19 TwinsUK is also supported by the Chronic Disease Research Foundation and Zoe Global Ltd. The funders had no role in study design, data collection and analysis, decision to publish, or preparation of the manuscript. Generation Scotland received core support from the Chief Scientist Office of the Scottish Government Health Directorates [CZD/16/6] and the Scottish Funding Council [HR03006]. Genotyping of the GS:SFHS samples was carried out by the Genetics Core Laboratory at the Wellcome Trust Clinical Research Facility, Edinburgh, Scotland and was funded by the Medical Research Council UK and the Wellcome Trust (Wellcome Trust Strategic Award "STratifying Resilience and Depression Longitudinally" (STRADL) Reference 104036/Z/14/Z). Generation Scotland is funded by the Wellcome Trust (216767/Z/19/Z) and (221574/Z/20/Z). Born in Bradford (BiB) receives core infrastructure funding from the Wellcome Trust (WT101597MA), and a joint grant from the UK Medical Research Council (MRC) and UK Economic and Social Science Research Council (ESRC) (MR/N024397/1), the British Heart Foundation (BHF) (CS/16/4/32482), and The Health Foundation COVID-19 award (2301201). The National Institute for Health Research Yorkshire and Humber Applied Research Collaboration (ARC) (NIHR200166), and Clinical Research Network both provide support for BiB research. Born in Bradford is only possible because of the enthusiasm and commitment of the children and parents in BiB. We are grateful to all the participants, health professionals, schools and researchers who have made Born in Bradford happen. OpenSAFELY is jointly funded by UKRI, NIHR and Asthma UK-BLF [COV0076; MR/V015737/] and the Longitudinal Health and Wellbeing strand of the National Core Studies programme. E.M.I.S. and T.P.P. provided technical expertise and infrastructure within their data environments pro bono in the context of a national emergency. The OpenSAFELY software platform is supported by a Wellcome Discretionary Award. B.G.'s work on clinical informatics is supported by the NIHR Oxford Biomedical Research Centre and the NIHR Applied Research Collaboration Oxford and Thames Valley. Funders had no role in the study design, collection, analysis, and interpretation of data; in the writing of the report; and in the decision to submit the article for publication. The views expressed are those of the authors and not necessarily those of the NIHR, NHS England, Public Health England or the Department of Health and Social Care. N.J.T. is a Wellcome Trust Investigator (202802/Z/16/Z), is the PI of the Avon Longitudinal Study of Parents and Children (MRC & WT 217065/Z/19/Z), is supported by the University of Bristol NIHR Biomedical Research Centre, the MRC Integrative Epidemiology Unit (MC_UU_00011/1) and works within the CRUK Integrative Cancer Epidemiology Programme (C18281/A29019). S.V.K. acknowledges funding from a NRS Senior Clinical Fellowship (SCAF/15/02), the Medical Research Council (MC_UU_00022/2) and the Scottish Government Chief Scientist Office (SPHSU17). A.S.F.K. acknowledges funding from the ESRC (ES/V011650/1). E.J.T. acknowledges funding from the Wellcome Trust (WT212904/Z/18/Z). R.M. acknowledges support from the Elizabeth Blackwell Institute for Health Research, University of Bristol, and the Wellcome Trust Institutional Strategic Support Fund (204813/Z/16/Z). G.B.P. acknowledges funding from the Economic and Social Research Council (ES/V012789/1). C.L.N. acknowledges funding from the Medical Research Council (MR/R024774/1). K.T. works in a Unit that is supported by the University of Bristol and UK Medical Research Council (MC_UU_00011/3). D.M.W. is supported by funding from UK Medical Research Council (MC_PC_20030). N.C. is supported by funding from the UK Medical Research Council (MC_UU_00019/2). We would also like to acknowledge the following individuals: Generation Scotland: Drew Altschul, Chloe Fawns-Ritchie, Archie Campbell, Robin Flaig; ALSPAC: Daniel J Smith; Understanding Society: Michaela Benzeval; TwinsUK: Deborah Hart, María Paz García, Rachel Horsfall; Centre for Longitudinal Studies: Matt Brown, Lisa Calderwood, Emla Fitzsimons, Alissa Goodman, Aida Sanchez; Born in Bradford: John Wright, Dan Mason.

## Author contributions

N.J.T., N.C. and C.J.S. conceptualised the study and design. E.J.T., D.M.W., A.J.W., R.J.S., K.T., C.T.R., N.J.T., N.C. and C.J.S. designed the methodology. E.J.T., D.M.W., A.J.W., R.E.M., C.L.N., T.C.Y., C.F.H. and A.S.F.K. conducted the formal analysis. E.J.T., D.M.W., A.J.W., R.E.M., C.L.N., T.C.Y., C.F.H., G.D., R.C.E.B., B.H., M.J.G., B.D., K.J.D., OpenSAFELY Collaborative, K.N., and A.S.F.K. were responsible for data curation. E.J.T., D.M.W., A.J.W. and C.J.S. wrote the original draft of the manuscript. E.J.T., D.M.W., A.J.W., R.E.M., C.L.N., T.C.Y., C.F.H., A.S.F.K., R.J.S., G.D., R.C.E.B., K.N., B.H., M.J.G., K.J.D., E.L.D., D.M.W., OpenSAFELY Collaborative, A.S., D.J.P., R.R.C.M., L.T., B.G., P.P., G.B.P., S.V.K., K.T., N.J.T., N.C., and C.J.S. contributed to critical revision of the manuscript. D.M.W. and A.J.W. contributed to data visualisation. The project was supervised by N.J.T., N.C. and C.J.S. Funding was acquired by S.V.K., D.J.P., A.S., B.G., P.P., G.B.P., R.J.S., N.J.T., N.C. and C.J.S.

## Competing interests

No competing interests were declared by E.J.T., D.M.W., A.J.W., R.E.M., C.L.N., T.C.Y., C.F.H., A.S.F.K., R.J.S., G.D., R.C.E.B., K.N. B.H., M.J.G., B.D., K.J.D., E.L.D., F.M.K.W., A.S., L.T., B.G., P.P., G.B.P., K.T., C.T.R., N.J.T., N.C., C.J.S. B.G. has received research funding from the Laura and John Arnold Foundation, the NHS National Institute for Health Research (NIHR), the NIHR School of Primary Care Research, the NIHR Oxford Biomedical Research Centre, the Mohn-Westlake Foundation, NIHR Applied Research Collaboration Oxford and Thames Valley, the Wellcome Trust, the Good Thinking Foundation, Health Data Research UK (HDRUK), the Health Foundation, and the World Health Organisation; he also receives personal income from speaking and writing for lay audiences on the misuse of science. S.V.K. is a member of the Scientific Advisory Group on Emergencies subgroup on ethnicity and COVID-19 and is co-chair of the Scottish Government's Ethnicity Reference Group on COVID-19. N.C. serves on a data safety monitoring board for trials sponsored by AstraZeneca. C.J.S. is an academic lead on KCL Zoe Global Ltd. COVID symptoms study. The remaining authors declare no competing interests.

## Additional information

## OpenSAFELY Collaborative

Alex J. Walker[4], Brian MacKenna[4], Peter Inglesby[4], Christopher T. Rentsch[15], Helen J. Curtis[4], Caroline E. Morton[4], Jessica Morley[4], Amir Mehrkar[4], Seb Bacon[4], George Hickman[4], Chris Bates[18], Richard Croker[4], David Evans[4], Tom Ward[4], Jonathan Cockburn[18], Simon Davy[4], Krishnan Bhaskaran[15], Anna Schultze[15], Elizabeth J. Williamson[15], William J. Hulme[4], Helen I. McDonald[15], Laurie Tomlinson[15], Rohini Mathur[15], Rosalind M. Eggo[15], Kevin Wing[15], Angel Y. S. Wong[15], Harriet Forbes[15], John Tazare[15], John Parry[18], Frank Hester[18], Sam Harper[18], Ian J. Douglas[15], Stephen J. W. Evans[15], Liam Smeeth[15] & Ben Goldacre[4]

[15]Electronic Health Records Research Group, Faculty of Epidemiology and Population Health, London School of Hygiene & Tropical Medicine, London, UK. [16]VA Connecticut Healthcare System, West Haven, CT, USA. [17]Department of Ageing and Health, Guys and St Thomas's NHS Foundation Trust, London, UK. [18]TPP, TPP House, 4159 Low Lane, Horsforth, Leeds LS48 5PX, UK.

