## [Peer Review File · Nature Communications]

Long COVID burden and risk factors in 10 UK longitudinal studies and electronic health recordsREVIEWER COMMENTS

Reviewer #1 (Remarks to the Author):

The authors aim to study the frequency of and risk factors for long COVID using data from over 10,000 UK longitudinal study (LS) and EHRs who were assigned a long COVID code. Multivariate logistic regression with random effects was applied to the data with pre-set of covariates representing sociodemographic and pre-pandemic health risk factors (age, sex, ethnicity, socioeconomic factors, smoking, general and mental health, overweight/obesity, diabetes, hypertension, hypercholesterolemia, and asthma).

- What are the noteworthy results?

Age was linearly associated with long COVID between 20 and 70.

Being female, having poor pre-pandemic mental health, poor general health, asthma, and obesity were associated with long covid consistently among the 2 data types (despite variability in frequency of long covid across the 2 data), but the ORs were generally below 1.4.

Non-white ethnic minority groups had lower risk of long covid.

- Will the work be of significance to the field and related fields? How does it compare to the established literature? If the work is not original, please provide relevant references.

The results are interesting but do not provide very significant findings to the field. The methods are standard.

- Does the work support the conclusions and claims, or is additional evidence needed?

Mostly. Please see the comment on methods.

- Are there any flaws in the data analysis, interpretation and conclusions? - Do these prohibit publication or require revision?

No.

- Is the methodology sound? Does the work meet the expected standards in your field?

Definition of covid positive is different in the 2 data (self-reported in LS and positive test or diagnosis code in EHR). Similarly, the long COVID definitions are different (in LS symptoms lasting 4+ weeks, combining OSC and PCS, symptoms lasting 0-4 weeks as reference). 12+ weeks was also considered as a primary outcome. In EHR, it was the long COVID codes (a list of 15 UK SNOMED codes, categorized as diagnostic (2 codes), referral (3) and assessment (10) codes) from primary care records. Again, similar thing with the covariates. This is a strength here as the authors only emphasize on the results that are consistent.

However, there is an important question mark: In the LS data, the authors seem to have evaluated the NICE criteria, which require that symptoms are "not explained by an alternative diagnosis." But is the EHR long covid assumed to be valid? A verification that the NICE criteria is still met will ensure that the outcome is indeed reliable.

There is also the question of whether symptoms lasting 4+ weeks should be considered as long COVID. Focus on 12+ weeks is preferable.

Finally, the sensitivity of self-reported COVID needs to be evaluated within the LS data.

- Is there enough detail provided in the methods for the work to be reproduced?

Yes

minor comment:

OSC and PCS aren't defined.

Reviewer #2 (Remarks to the Author):

This study assesses potential predictors of long COVID, defined two ways, with a sample of 10 longitudinal studies (LS) and electronic health records (EHR). There are several noteworthy findings consistent with previous studies on long COVID, including associations between long COVID and age, female sex, and pre-existing asthma and psychological disorders. Although findings are largely consistent between the longitudinal studies and EHR, there are several conflicting findings, including the relationship between increasing age and long COVID and area-level deprivation and long COVID. The main strength of the study is the sample it draws from, although neither sample (LS or EHR) is truly population based (in terms of being representative of the general UK population) and results should not be interpreted as such. Additional suggestions and clarifying questions are listed below.

Abstract

- Typo in line 50: Should read "association with age, sex, ethnicity..."
- Line 54 refers to each Long COVID outcome, but these outcomes are not defined in the measures section above. If space is an issue, there is no need to repeat all the measures currently listed in the measures section in the analysis section as well.
- The EHR component as described in the abstract is unclear. Were the LS participants all patients and therefore all had electronic health records? Are the EHR records at one point in time only? How were the EHR results analyzed? In line 57, you state that random effects meta-analysis was used to combine results across the 10 cohorts and EHR but results are presented separately in the next section.

Methods

- Line 162: Please describe the Index of Multiple Deprivation and provide a reference.
- Lines 170-171: Was there a reason anxiety was not included in the list of pre-existing mental health diagnoses? Does it have a separate diagnostic code?
- Lines 177-182: Why did the authors choose to examine overweight vs. not in the LS and obese vs. not in the EHR? Consistency would be helpful for interpretation and a 3 category BMI variable (underweight/normal weight, overweight, obese) is preferable.
- Line 196 refers to the COVID outcomes as Long COVID and PCS but previously the authors refer to both outcomes (OSC or PCS) as Long COVID. Please use consistent naming throughout for clarity.
- The approach to modeling age across the LS and EHR is unclear. The authors state they adjusted for age as a continuous variable in line 198. Previous studies have found a non-linear association between age and Long COVID, with a slight decline in risk for the oldest age group. The next paragraph describes age modeled in different ways. Was continuous age used as a control only? In line 227 the authors state that age was adjusted for as a categorical variable in the EHR analysis. Was the LS sample restricted to age 70 or less? Age should be controlled for in whatever functional form is most appropriate based on the relationship between age and long COVID in these studies but consistency between the LS and EHR analyses is also important when possible. It seems like categorical age would show both the linear association seen in LS and the inverted U shape seen in EHR and would allow comparison across study types.
- What is the date range of COVID diagnosis for both the LS sample and EHR? And the dates of long COVID diagnosis for the EHR?

Results

- Line 289: The phrase "compared to" should be removed
- The results section goes back and forth on using OSC and PCS vs. 4+ weeks and 12+ weeks. It would be easier for the reader to stick with 4+ and 12+ weeks throughout the entire paper.

Discussion

- Line 361: This line should be updated to say that the odds of long COVID were 50% higher in women than men, since the ORs are not measuring prevalence or risk.
- Lines 373-380: This study certainly adds evidence on previously conflicting risk factors for long COVID but I don't think it's appropriate to say it resolves these issues (lines 375 and 379).
- Lines 394-397: I don't understand the assertion that the complete case analysis is unbiased given the use of logistic regression for a binary outcome. Issues like selection bias and information bias cannot be adequately addressed at the analytic stage.

REVIEWER COMMENTS

Reviewer #1 (Remarks to the Author):

The authors aim to study the frequency of and risk factors for long COVID using data from over 10,000 UK longitudinal study (LS) and EHRs who were assigned a long COVID code. Multivariate logistic regression with random effects was applied to the data with pre-set of covariates representing sociodemographic and pre-pandemic health risk factors (age, sex, ethnicity, socioeconomic factors, smoking, general and mental health, overweight/obesity, diabetes, hypertension, hypercholesterolemia, and asthma).

- What are the noteworthy results?

Age was linearly associated with long COVID between 20 and 70.

Being female, having poor pre-pandemic mental health, poor general health, asthma, and obesity were associated with long covid consistently among the 2 data types (despite variability in frequency of long covid across the 2 data), but the ORs were generally below 1.4.

Non-white ethnic minority groups had lower risk of long covid.

- Will the work be of significance to the field and related fields? How does it compare to the established literature? If the work is not original, please provide relevant references.

The results are interesting but do not provide very significant findings to the field. The methods are standard.

- Does the work support the conclusions and claims, or is additional evidence needed?

Mostly. Please see the comment on methods.

- Are there any flaws in the data analysis, interpretation and conclusions? - Do these prohibit publication or require revision?

No.

- Is the methodology sound? Does the work meet the expected standards in your field?

Definition of covid positive is different in the 2 data (self-reported in LS and positive test or diagnosis code in EHR). Similarly, the long COVID definitions are different (in LS symptoms lasting 4+ weeks, combining OSC and PCS, symptoms lasting 0-4 weeks as reference). 12+ weeks was also considered as a primary outcome. In EHR, it was the long COVID codes (a list of 15 UK SNOMED codes, categorized as diagnostic (2 codes), referral (3) and assessment (10) codes) from primary care records. Again, similar thing with the covariates. This is a strength here as the authors only emphasize on the results that are consistent.

However, there is an important question mark: In the LS data, the authors seem to have evaluated the NICE criteria, which require that symptoms are "not explained by an alternative diagnosis." But is the EHR long covid assumed to be valid? A verification that the NICE criteria is still met will ensure that the outcome is indeed reliable.

RESPONSE: Thank you for highlighting this potential difference in the long COVID definitions. In the EHR data we are inherently reliant on the individual judgement of GPs who enter long COVID diagnostic codes into the patient record. However, given that the SNOMED codes used are very closely modelled on the NICE guidelines (<https://www.nice.org.uk/guidance/ng188>), we think that it

is reasonable to assume that the positive predictive value of having a long COVID SNOMED code is high.

There is also the question of whether symptoms lasting 4+ weeks should be considered as long COVID. Focus on 12+ weeks is preferable.

RESPONSE: NICE guidelines specify symptoms longer than 4 weeks are classified as long COVID (also see: <https://www.nice.org.uk/guidance/ng188>)

Finally, the sensitivity of self-reported COVID needs to be evaluated within the LS data.

RESPONSE: Thank you for raising this. Within two cohorts with comprehensive antibody data, we have evaluated symptom duration and risk factors which are in supplementary figure 11-14. These show some similar effect sizes and direction of effect as the self-reported results for some of the predictors (i.e., female sex), albeit some were non-significant (i.e. psychological distress) and changed direction (i.e. asthma). However, these results should be treated with caution owing to the small sample sizes the synthesised effects are based on. In addition, in the discussion we have added the following sentence:

“Sensitivity analysis of those with positive PCR/antibody data showed some inconstancies in direction of effect estimates. However, as results should be treated with caution due to the small sample sizes included.”

- Is there enough detail provided in the methods for the work to be reproduced?

Yes

minor comment:

OSC and PCS aren't defined

RESPONSE: Thank you for raising this issue, as we appreciate these definitions may be confusing. We previously defined OSC and PCS in the introduction. However, to aid clarity, we have removed all references to ongoing symptomatic COVID-19 (OSC) and post-COVID-19-syndrome (PCS) and refer only to the duration of symptoms in weeks.

Reviewer #2 (Remarks to the Author):

This study assesses potential predictors of long COVID, defined two ways, with a sample of 10 longitudinal studies (LS) and electronic health records (EHR). There are several noteworthy findings consistent with previous studies on long COVID, including associations between long COVID and age, female sex, and pre-existing asthma and psychological disorders. Although findings are largely consistent between the longitudinal studies and EHR, there are several conflicting findings, including the relationship between increasing age and long COVID and area-level deprivation and long COVID. The main strength of the study is the sample it draws from, although neither sample (LS or EHR) is truly population based (in terms of being representative of the general UK population) and results should not be interpreted as such. Additional suggestions and clarifying questions are listed below.

Abstract –

- Typo in line 50: Should read “association with age, sex, ethnicity...”

RESPONSE: This typo has now been corrected

- Line 54 refers to each Long COVID outcome, but these outcomes are not defined in the measures section above. If space is an issue, there is no need to repeat all the measures currently listed in the measures section in the analysis section as well.

RESPONSE: We have now defined the long COVID outcome measures in the ‘measures’ section of the abstract.

- The EHR component as described in the abstract is unclear. Were the LS participants all patients and therefore all had electronic health records? Are the EHR records at one point in time only? How were the EHR results analysed? In line 57, you state that random effects meta-analysis was used to combine results across the 10 cohorts and EHR but results are presented separately in the next section.

RESPONSE: Thank you for this comment. Random effects meta-analysis was used to combine results across the 10 longitudinal studies. Data from EHRs was analysed separately and these effect sizes were not included in the meta-analysis. We have now made this clearer in the abstract, as follows:

“Univariate logistic regression was used to assess the association between each long COVID outcome and each predictor (adjusted for age and sex for all risk factors except age and sex). Random effects meta-analysis was used to combine results across the 10 LS. Across LS, inverse probability weighting (IPW) was used to examine sensitivity to selection and index event bias. Logistic regression was run separately on EHR data and is presented in parallel to the meta-analysed LS pooled data.”

Methods

- Line 162: Please describe the Index of Multiple Deprivation and provide a reference.

RESPONSE: This has now been described in more detail and referenced accordingly:

“Index of Multiple Deprivation (IMD; divided into quintiles with 1 representing the most deprived and 5 representing the least deprived). Area-level SES was measured using the Index of Multiple Deprivation 2019 (IMD), a composite of different domains including area-level income, employment,

education access and crime, for the postcode where a participant lived at the time of sample collection.(10)"

Reference: ONS Postcode Directory (Latest) Centroids | Open Geography Portal. Accessed December 8, 2021. <https://geoportal.statistics.gov.uk/datasets/ons::ons-postcode-directory-latest-centroids/about>

- Lines 170-171: Was there a reason anxiety was not included in the list of pre-existing mental health diagnoses? Does it have a separate diagnostic code?

RESPONSE: Thank you for this comment. As the majority of pre-pandemic mental health measures used by the cohorts are either measures of general psychological distress (USOC: General Health Questionnaire-12) or specific measures of depression (i.e., BCS70: Malaise Inventory; ALSPAC: Short Mood and Feelings Questionnaire (SMFQ)), in order to increase consistency between EHR data and LS, only codes reflecting a depression assignment were used.

- Lines 177-182: Why did the authors choose to examine overweight vs. not in the LS and obese vs. not in the EHR? Consistency would be helpful for interpretation and a 3 category BMI variable (underweight/normal weight, overweight, obese) is preferable.

Thank you for bringing this up. It is a limitation that different metrics for body weight have been used. This is because, within LS numbers in the obese category were relatively small, e.g., 8.9% for TwinsUK, whereas in EHR codes were defined using obesity because ascertaining when someone is obese is likely to be more reliable in EHR than it is to ascertain when someone is overweight. The logic here is that obesity is more likely to affect patient health and is therefore more likely to be recorded by a clinician. Nevertheless, finding reproducibility of association despite different definition indicates strengthens our confidence in this finding.

- Line 196 refers to the COVID outcomes as Long COVID and PCS but previously the authors refer to both outcomes (OSC or PCS) as Long COVID. Please use consistent naming throughout for clarity. –

RESPONSE: Thank you for raising the inconsistencies in our long COVID outcome labels. As noted earlier, for clarity we are now only referring to the long COVID categories defined by NICE, along with our outcome variables based on these definitions, in terms of weeks of symptom duration. All uses of OSC and PCS have been removed.

- The approach to modelling age across the LS and EHR is unclear. The authors state they adjusted for age as a continuous variable in line 198. Previous studies have found a non-linear association between age and Long COVID, with a slight decline in risk for the oldest age group. The next paragraph describes age modelled in different ways. Was continuous age used as a control only? In line 227 the authors state that age was adjusted for as a categorical variable in the EHR analysis. Was the LS sample restricted to age 70 or less? Age should be controlled for in whatever functional form is most appropriate based on the relationship between age and long COVID in these studies but consistency between the LS and EHR analyses is also important when possible. It seems like categorical age would show both the linear association seen in LS and the inverted U shape seen in EHR and would allow comparison across study types.

RESPONSE: The reviewer raises a good point about the appropriate handling of age distributions, both as a covariate and its own direct relationship with long COVID risk. We appreciate that there is possibility for confusion here.

The first paragraph in the '*Statistical analysis: LS*' section refers to age being modelled as a covariate when testing other exposure – long COVID associations. This was appropriate because *within* studies, the age ranges were quite limited, as shown by data in Table 1. While we did not restrict LS samples to individuals under 70 years of age, these comprised the vast majority of samples (and exclusively so for several of the LS). Within studies, we do not expect the age – long COVID relationship to display the same extent of non-linearity -- i.e. an inverse U shape – as seen across individuals in samples with very large ranges spanning older age groups (as we observe within EHR data). Thus, while we cannot entirely exclude some small degree of non-linearity age as an adjustment with long COVID in some of the age-heterogeneous LS, this is likely to be modest at most and we are confident that our approach is the most appropriate option when considering age as an adjustment for other risk factor- long COVID associations.

The second paragraph in the '*Statistical analysis: LS*' section addresses the direct relationship of age with long COVID. Given the different age structures of LS, we could not do this with exactly the same methods across all LS or between all LS and the EHR data, hence the dual approach.

We have added some additional text to these two paragraphs in the revised manuscript to help clarify these matters.

- What is the date range of COVID diagnosis for both the LS sample and EHR? And the dates of long COVID diagnosis for the EHR?

RESPONSE: Thank you for this comment. We have added this sentence to the measures section of the methods for LS: "*All data used to derive these outcomes were collected between April-November 2020.*" And for EHR "*The outcome was measured between the study start date (2020-02-01) and the end date (2021-05-09).*"

Results

- Line 289: The phrase "compared to" should be removed

RESPONSE: This has now been corrected

- The results section goes back and forth on using OSC and PCS vs. 4+ weeks and 12+ weeks. It would be easier for the reader to stick with 4+ and 12+ weeks throughout the entire paper.

RESPONSE: As noted earlier, it is clear that the use of OSC and PCS definitions introduce some confusion, so we only refer to symptom durations in terms of number of weeks in the revised manuscript.

Discussion

- Line 361: This line should be updated to say that the odds of long COVID were 50% higher in women than men, since the ORs are not measuring prevalence or risk.

RESPONSE: We have now added '*the odds of*' to this sentence, which now reads: *The findings that the odds of long COVID was 50% higher in women than men is consistent with reports from most...*

- Lines 373-380: This study certainly adds evidence on previously conflicting risk factors for long COVID but I don't think it's appropriate to say it resolves these issues (lines 375 and 379).

RESPONSE: Thank you highlighting this. We have now removed the term 'resolves' so the sentence now reads: "*Excess risk of long COVID in association with asthma across cohorts and primary care records combats previous resolves conflicting and limited findings*" ...

- Lines 394-397: I don't understand the assertion that the complete case analysis is unbiased given the use of logistic regression for a binary outcome. Issues like selection bias and information bias cannot be adequately addressed at the analytic stage.

RESPONSE: Thank you for highlighting this. We have now removed this sentence from the discussion and added how we combatted potential index response bias in the cohorts through utilising inverse probability weights for the probability of getting COVID-19.

REVIEWER COMMENTS

Reviewer #1 (Remarks to the Author):

The authors provided responses that to some extent satisfy some of the comments raised on the original manuscript. However, since the edits were not marked, it is difficult to assess where in the manuscript was edited. Below are the main remaining concerns:

1-There has been a number of articles out now on long covid. The references cited in the manuscript are fairly limited. It is necessary that the authors provide a short overview of what is now known in the literature on long covid and identify which of the findings is novel or confirmatory compared to the published knowledge base.

2-On the comment related to the validity of the EHR long covid records, the authors' response refer to their reliance on the "individual judgement of GPs who enter long COVID diagnostic codes into the patient record" and that "SNOMED codes used are very closely modelled on the NICE guidelines..." Regardless of these assumption that seem fair, as a scientific experiment, some level of the evaluation of the HER long covid diagnosis is needed to ensure similarities across the outcomes from the 2 data sources.

3-In response to "the sensitivity of self-reported COVID needs to be evaluated within the LS data" the authors have added two new sentences to the discussion: "Sensitivity analysis of those with positive PCR/antibody data showed some inconstancies in direction of effect estimates. However, as results should be treated with caution due to the small sample sizes included." This is big uncertainty that is not satisfactory. Similar to the previous comments, a sensitivity analysis needs to be performed on the self-reported COVID within the LS data. Essentially, how many of the self-reported really had COVID?. The authors did some sensitivity analyses to show that it effects sizes and directions are similar, but alternatively one could perform analysis assuming different %s of correct self-reported infections.

Reviewer #2 (Remarks to the Author):

Thank you for your attention to my previous comments. There were several instances where you clarified information for me but did not fully update information in the manuscript (or at least did not make it clear that you did so based on your rebuttal). Please add clarifying text to the manuscript itself so readers will understand the following points:

- Reasons for using different overweight/obesity categorical variables across LS and EHR analyses
- Since age is such as important predictor of COVID (and potentially long COVID) outcomes, I think more clarity is needed on the different approaches to modeling age in the manuscript itself. At a minimum, the authors should include the age categories used in the main text, rather than referring to supplemental figures. Additionally, it is still unclear how age was treated in the analysis described in lines 213-215.

REVIEWER COMMENTS

Reviewer #1 (Remarks to the Author):

The authors provided responses that to some extent satisfy some of the comments raised on the original manuscript. However, since the edits were not marked, it is difficult to assess where in the manuscript was edited. Below are the main remaining concerns:

1-There has been a number of articles out now on long covid. The references cited in the manuscript are fairly limited. It is necessary that the authors provide a short overview of what is now known in the literature on long covid and identify which of the findings is novel or confirmatory compared to the published knowledge base.

RESPONSE: Thank you for highlighting the need to update the literature review in this paper. We have now added the following sentence and references to the introduction:

“Emerging evidence indicates risk factors for long COVID including demographic characteristics,¹⁻³, comorbidities,³ and immunological response.¹ However, existing studies often rely on cross-sectional data from small sample. Accurate risk estimates require large generalisable samples with comprehensive measures of pre-pandemic characteristics.⁴”

1. Cervia C, Zurbuchen Y, Taeschler P, et al. Immunoglobulin signature predicts risk of post-acute COVID-19 syndrome. *Nat Commun.* 2022;13(1). doi:10.1038/s41467-021-27797-1
2. Mortimer RC. Risk factors associated with development and persistence of long COVID. *Christ Ethics.* Published online 2021:3-3. doi:10.4324/9780203534595-2
3. Sudre CH, Murray B, Varsavsky T, et al. Attributes and predictors of long COVID. *Nat Med.* 2021;27(April). doi:10.1038/s41591-021-01292-y
4. Amin-Chowdhury Z, Ladhani SN. Causation or confounding: why controls are critical for characterizing long COVID. *Nat Med.* 2021;27(7):1126-1127. doi:10.1038/s41591-021-01395-6

2-On the comment related to the validity of the EHR long covid records, the authors' response refer to their reliance on the "individual judgement of GPs who enter long COVID diagnostic codes into the patient record" and that "SNOMED codes used are very closely modelled on the NICE guidelines..." Regardless of these assumption that seem fair, as a scientific experiment, some level of the evaluation of the EHR long covid diagnosis is needed to ensure similarities across the outcomes from the 2 data sources.

RESPONSE: Thank you for this comment. We have raised this query with Dr Gail Allsopp (GP and the Clinical Policy lead for the Royal College of General Practitioners) who responded with a helpful insight as follows: *“The diagnosis of Ongoing symptomatic COVID-19 and Post COVID- 19 syndrome are clinical diagnoses made on the basis of evidence of a preceding COVID-19 infection (clinical or test diagnosis) and exclusion of other causes of their symptoms as per the NICE/SIGN/RCGP guidance. The codes align directly with the diagnosis as they were developed after the case definition was published in December 2020. Primary care teams will only code something in the patient notes when it is a definitive diagnosis as it stays with the person for their whole life within their care record. There is no ‘check’ on whether a clinician has coded correctly for any condition, but as with any patient notes, clinicians are bound by their professional registrations to ensure the coding is correct.”*

3-In response to “the sensitivity of self-reported COVID needs to be evaluated within the LS data” the authors have added two new sentences to the discussion: “Sensitivity analysis of those with positive PCR/antibody data showed some inconsistencies in direction of effect estimates. However, as results should be treated with caution due to the small sample sizes included.” This is big uncertainty that is not satisfactory. Similar to the previous comments, a sensitivity analysis needs to be performed on the self-reported COVID within the LS data. Essentially, how many of the self-reported really had COVID? The authors did some sensitivity analyses to show that its effects sizes and directions are similar, but alternatively one could perform analysis assuming different %s of correct self-reported infections.

RESPONSE: We agree with the reviewer that it is a major limitation of our research (and all similar large-scale analyses that we are aware of) to lack test confirmation of SARS-Cov-2 exposure in all self-reported cases. It is beyond the scope of this work to run further analyses and simulations to quantify the extent of bias from misclassification and its impact on potential risk factor - long COVID associations – efforts that we think would warrant its own publication and would hinder the timely publication of the current data further. However, we have expanded the commentary on this issue and added reference to previous data to the discussion to emphasise this limitation further:

“Finally, not all studies had test confirmation of COVID-19 status, and some individuals may have misattributed persistent symptoms to other conditions. From past analyses to establish case definitions in two of the samples (ALSPAC G0 and G1), 25.8 and 32.2% of self-reported cases could be verified against PCR results from linked national testing and/or serology, respectively. Though this implies that there may be many self-reported COVID-19 instances in the samples prone to misclassification, there are issues with test confirmation that mean true misclassification may be much lower (e.g., limited surveillance with PCR testing, imperfect sensitivity of both PCR and antibody tests, and waning antibody titres leading to seroreversion over time). The impact of bias from misclassification on the risk factor associations with long COVID is unclear. Sensitivity analysis of those with positive PCR/antibody data showed some inconsistencies in directions of effect estimates associations. However, as these results should be treated with caution interpreted cautiously due to the small sample sizes included, and further collections of test data on large-scale LPS will be required .to augment the number of confirmed cases for similar analyses in future.”

Reviewer #2 (Remarks to the Author):

Thank you for your attention to my previous comments. There were several instances where you clarified information for me but did not fully update information in the manuscript (or at least did not make it clear that you did so based on your rebuttal). Please add clarifying text to the manuscript itself so readers will understand the following points:

- Reasons for using different overweight/obesity categorical variables across LS and EHR analyses

RESPONSE: Thank you for highlighting the need to add this to the manuscript. Following our last response on this point we have now added an additional sentence to the ‘Overweight and obesity’ sub-section:

“A BMI of >25 was used in LS as the percentage of those in the obese category (i.e., BMI >30) was relatively small, e.g., 8.9% for TwinsUK, whereas EHR obesity codes were used as these are more reliable and valid indicators of having obesity in general practice.”

- Since age is such an important predictor of COVID (and potentially long COVID) outcomes, I think more clarity is needed on the different approaches to modelling age in the manuscript itself. At a minimum, the authors should include the age categories used in the main text, rather than referring to supplemental figures. Additionally, it is still unclear how age was treated in the analysis described in lines 213-215.

RESPONSE: Thank you for this comment. We have now added a sentence to lines 213-215

“First, in age-heterogeneous samples we compared long COVID risk within pre-defined age categories, comparing 45-69 and 70+ to 18-44 (reference category) in three cohorts (USOC, TwinsUK and GS), and 55-59 and 60-76 to 45-54 (reference category) in one cohort (ALSPAC G0).”